# Human-Borne Pathogens: Are They Threatening Wild Great Ape Populations?

**DOI:** 10.3390/vetsci9070356

**Published:** 2022-07-13

**Authors:** Pamela C. Köster, Juan Lapuente, Israel Cruz, David Carmena, Francisco Ponce-Gordo

**Affiliations:** 1Parasitology Reference and Research Laboratory, Spanish National Centre for Microbiology, Health Institute Carlos III, Majadahonda, 28220 Madrid, Spain; pamelkaster@yahoo.es; 2Comoé Chimpanzee Conservation Project (CCCP) Comoé N.P., Kakpin, Côte d’Ivoire; juanlapuente@yahoo.com; 3National School of Public Health, Health Institute Carlos III, 28029 Madrid, Spain; cruzi@isciii.es; 4Center for Biomedical Research Network (CIBER) in Infectious Diseases, Health Institute Carlos III, Majadahonda, 28220 Madrid, Spain; 5Department of Microbiology and Parasitology, Faculty of Pharmacy, Complutense University of Madrid, 28040 Madrid, Spain

**Keywords:** virus, bacteria, parasites, fungi, anthropic activities, zoonosis, disease transmission, animal diversity, conservation, infectious diseases

## Abstract

**Simple Summary:**

Human-driven activities, including agriculture, forestry, and mining, are destroying the natural habitats of wild great ape (bonobo, chimpanzee, gorilla, and orangutan) populations in Africa and Southeast Asia. The reduction in and fragmentation of wild great ape environments lead to (i) a decrease in population numbers, (ii) the isolation of current populations, and (iii) increased exposure to humans and their livestock. Consequently, the spatial overlap between humans and wild great apes might facilitate the transmission of infectious agents between them. Historically, animal-to-human pathogen transmission has attracted most of the attention of researchers and public health authorities. Only in recent years has the human-to-animal transmission pathway acquired notoriety, mainly due to conservation concerns. In this review, we examine and appraise literature-based evidence reporting wild great ape infections with viral, bacterial, parasitic, and fungal pathogens of potential anthropic nature. We select and further discuss two viral (Human Metapneumovirus and Respiratory Syncytial Virus), one bacterial (diarrhoeagenic *Escherichia coli*), and two parasitic (*Cryptosporidium* spp. and *Giardia duodenalis*) pathogens causing infections in wild great ape populations for which a human origin is most likely. Gaps in knowledge and future research directions are also identified.

**Abstract:**

Climate change and anthropic activities are the two main factors explaining wild great ape habitat reduction and population decline. The extent to which human-borne infectious diseases are contributing to this trend is still poorly understood. This is due to insufficient or fragmented knowledge on the abundance and distribution of current wild great ape populations, the difficulty obtaining optimal biological samples for diagnostic testing, and the scarcity of pathogen typing data of sufficient quality. This review summarises current information on the most clinically relevant pathogens of viral, bacterial, parasitic, and fungal nature for which transmission from humans to wild great apes is suspected. After appraising the robustness of available epidemiological and/or molecular typing evidence, we attempt to categorise each pathogen according to its likelihood of truly being of human origin. We further discuss those agents for which anthroponotic transmission is more likely. These include two viral (Human Metapneumovirus and Respiratory Syncytial Virus), one bacterial (diarrhoeagenic *Escherichia coli*), and two parasitic (*Cryptosporidium* spp. and *Giardia duodenalis*) pathogens. Finally, we identify the main drawbacks impairing research on anthroponotic pathogen transmission in wild great apes and propose research lines that may contribute to bridging current knowledge gaps.

## 1. Introduction

Climate change and human interference are the two main reasons for great ape habitat decline [1,2]. The subsequent drastic drop in great ape populations has highlighted the urgent need to improve our knowledge on their actual situation. Bonobos (*Pan paniscus*) and chimpanzees (*Pan troglodytes*) are listed by the International Union for Conservation of Nature as endangered; eastern gorillas (*Gorilla beringei*), western gorillas (*Gorilla gorilla*), Sumatran orangutans (*Pongo abelii* and *Pongo tapanuliensis*), and Bornean orangutans (*Pongo pygmaeus*) are critically endangered [3,4]. All of them are experiencing marked declining population trends [5].

The decline in eastern and western gorilla populations over 20 years is estimated to have been more than 70% and 80%, respectively [4]. Although *P. troglodytes* is the most abundant and widespread of the great apes, and many populations exist in protected areas, the decline in their populations is expected to continue [6]. In fact, they are already extinct in 4 of their 25 distribution countries: Benin, Burkina Faso, Gambia, and Togo. It is estimated that at the beginning of the 20th century, the global chimpanzee population was around 1 million individuals. Today, it is estimated that there are between 172,000 and 300,000 chimpanzees left in the wild [7]. The size of the current population of the bonobo, which only inhabits the Democratic Republic of the Congo, is uncertain. Estimates based on the study of protected areas suggest a minimum population of 15,000 to 20,000 individuals [8,9]. The population decline over a three-generation (75-year) period from 2003 to 2078 is likely to exceed 50% [4]. Predictions based on current land use in Borneo and Sumatra estimate that of the approximately 13,846 individuals of *P. abelii* that exist today, 4000 could be lost by 2030, and by 2060, there could be an 81% decline in the population compared to the population in 1985 [1,4]. In addition, the population of *P. pygmaeus* was reduced by more than 60% between 1950 and 2010, and a further 22% decline is projected to occur between 2010 and 2025 [4]. The population of *P. tapanuliensis* is estimated to have been about 1489 individuals in 1985 and around 800 in 2017, when the species was first described [10], and is expected to be reduced by 83% over the course of three generations [4]. Current trends indicate that African ape populations will decline by a further 80 percent over the next 30 to 40 years [7].

The destruction of natural habitats due to the increasing demand for land for agriculture, forestry, mining activities, and human settlements has resulted in an increase in the interface between humans and wildlife. This spatial overlap facilitates the transmission of pathogens of both animal and human origin, with the latter posing an increasing threat to the survival of wild great ape populations [5,11,12]. Collaterally, the expansion of access routes to exploited forests has facilitated and intensified poaching. Corruption, lack of law enforcement, insufficient capacity and resources, and political instability, together with some outbreaks of infectious diseases such as the Ebolavirus [13], are also major threats to these populations [4].

Obtaining accurate data on the size of existing great ape populations in the wild is not feasible. Many populations were, and several are still, distributed outside protected areas in different countries, so it is difficult to obtain reliable statistics. Although fragmented and uneven, Appendix A attempts to provide an estimation of current great ape population sizes in the countries examined in this study, with special emphasis on identifying the numbers of habituated (see below) animals. Everything points to the fact that, in the coming decades, these ape populations that lack sufficient protection by the states will gradually disappear, leaving these species represented only in those countries and enclaves with greater conservation measures [14]. Some facts support this theory, as in the case of eastern gorillas. While *G. beringei* is experiencing a declining population trend, the Virunga Mountains gorilla subpopulation has shown a population increase [3,15,16]. The gorilla population that received only conventional conservation measures was shown to decline by 0.7% per year from 1967 to 2008 [15], while the population of habituated gorillas increased by 4.1%, a difference attributed to extreme conservation measures, human intervention, veterinary attention, and individual control [5].

Habituation consists of accustoming a community of primates to human presence through prolonged exposure to make these individuals, which are normally very elusive, indifferent to the presence of researchers or tourists [17]. Great ape habituation has its supporters and detractors. Constant observation tourism and/or ongoing research projects may have strong positive effects, such as (i) suppressing poaching around the studied or protected area, (ii) allowing more exhaustive veterinary control of the members of a group and improving the welfare of the habituated community as a whole, and (iii) attracting income and training opportunities for local inhabitants [5,18]. Thus, tourism based on the observation of great apes in their natural habitats has grown in recent decades and is often proposed as one of the best ways to guarantee the survival of great apes and their habitats in Africa and Southeast Asia [19,20]. In this review, when we talk about habituated great apes, we refer to wild animals, as opposed to the contraposition with the term “semi-free-living populations” used by other authors to refer to habituated animals [5].

Other authors have warned about the dangers involved in the habituation of wild great apes since tourists and researchers will always have some degree of impact on the environment [21], and close contact carries considerable risks of disease transmission [19,22,23]. Under this scenario, habituation may have the opposite of the desired effect, since a rising incidence of epidemics could result in decreasing primate populations [18,24,25,26]. The challenge of great ape habituation is to maximise the benefit of research and tourism while minimizing negative side effects [18]. Therefore, the decision to habituate a group of wild great apes must weigh costs and benefits [27,28,29].

Since the main routes of transmission of human diseases to apes are respiratory (aerosol) and faecal–oral, it is necessary to assess which human activities pose a greater risk to wild primate populations [17,19]. The risk of infection due to aerosol inhalation is directly proportional to the closeness of human–ape contact. Coughing, sneezing, and spitting can all project infectious aerosols several metres [30]. Therefore, it is not surprising that from 1968 to the present, outbreaks of respiratory diseases have been consistently reported in communities of habituated great apes [31]. Habituation may also increase the incidence of transmission of human pathogens, such as bacteria, viruses, and intestinal parasites, through the faecal–oral route [20]. Coprophagy, common in great apes, may enhance the faecal–oral transmission of intestinal pathogens [32].

Close and frequent contact with humans is not the only factor contributing to disease transmission to great apes. The shared use of habitats may be just as, or even more, important for the spreading of human-derived pathogens as close contact with people [33]. The fragmentation of natural habitats and, in some instances, habituation have been shown to induce behavioural changes in primates. For instance, primates are often seen roaming cultivated land, where the possibility of contracting human infections is greater because it is a humanised environment [19,20].

This review compiles the most clinically relevant pathogens affecting wild great apes for which a human origin was suspected. It should be noted that evidence unquestionably demonstrating anthroponotic transmission events is generally lacking because (i) studies detecting pathogens simultaneously in wild great ape and human populations sharing habitats are scarce, (ii) there are limited molecular data at the strain/genotype level, essential to ascertain the occurrence and directionality of zoonotic transmission events, (iii) anthroponotic origin is usually inferred from available (often partial or inconclusive) data, and (iv) evolution and host adaptation leading to lower pathogenicity may reduce the likelihood of human-borne pathogen transmission.

Table 1 shows the selected pathogen list according to their viral, bacterial, parasitic (microeukaryotes, helminths, and arthropods), and fungal nature. Information also includes the country where the study was conducted and the primate species affected, accounting for their habituated/non-habituated status. An attempt was also made to categorise (uncertain, unlikely, probable, or very probable) the likelihood that a given pathogen was truly of anthroponotic origin. To do so, we appraised the quality and robustness of the epidemiological and/or molecular typing evidence provided in each individual study considered in Table 1. Respiratory diseases of viral or bacterial origin were the most extensively studied. This is not surprising, considering that respiratory infections are the main cause of morbidity and mortality in wild chimpanzees and gorillas in many settings [34]. Of note, viruses that are relatively benign in humans (e.g., human rhinovirus C) can cause lethal outbreaks in ape populations, indicating poor host adaptation and a lack of resistance in apes [34]. Although not irrefutable, there is reasonable evidence that airborne virus transmission can occur between infected humans and wild great apes. This is particularly true for adenoviruses, coronavirus, and metapneumovirus. Human-borne viruses transmitted through the faecal–oral pathway (e.g., some adenoviruses) can also potentially infect non-human primates, primarily through environmental (water or green vegetables) contamination. Transmission through contact with body fluids (e.g., Ebola virus and monkeypox virus) seems far less likely (Table 1).

Several bacterial pathogens infecting wild great ape populations have been suspected of having a human origin based on their shared presence in both hosts. However, insufficient or absent microbiological data (e.g., *Salmonella* or *Streptococcus pneumoniae* serovars and human-adapted thermotolerant *Campylobacter* species) makes it difficult, if not impossible, to assess the extent to which anthroponotic bacterial transmission is taking part in natural habitats. The same is also true for *Escherichia coli* pathotype identification, for which full genotyping requires culturing, isolation, and molecular characterisation, including Sanger sequencing. These procedures have not always been fully conducted in the available literature (Table 1). The strongest evidence of human-derived bacterial infections in wild great apes comes from the detection of antimicrobial resistance markers (known to be circulating in human populations under antibiotic pressure) in bacterial strains isolated from individuals who did not receive antimicrobial treatment (see Section 3 below).

Among the microeukaryotic intestinal parasites with higher pathogenic potential, *Cryptosporidium* spp. and *Giardia duodenalis* are the species most frequently detected in wild great ape populations. However, no diarrhoeal outbreaks of cryptosporidiosis or giardiasis have been documented to date in these hosts, suggesting that subclinical infections with both pathogens are the norm rather than the exception. Although comparatively much less studied, the same seems to be true for fungal pathogens, for which the microsporidia *Enterocytozoon bieneusi* and *Encephalitozoon* spp. are the pathogens most investigated in wild great apes. Evidence of human-borne transmission can also be inferred for several microeukaryotic parasites (*Blastocystis* sp., *Cryptosporidium* spp., *Entamoeba histolytica*, and *G. duodenalis*) and fungi (microsporidia), for which well-known zoonotic species/genotypes have been simultaneously described in human and non-human primates sharing natural habitats. Unfortunately, the insufficient number of isolates fully characterised at the sub-genotype level does not allow the confirmation of such events. However, the fact that most of these pathogens can be faecal–orally transmitted suggests that environmental contamination with human faeces is a source of parasite and fungal infections in wild great apes.

To illustrate the impact of human-derived pathogens on wild great ape populations, we selected as examples those agents for which evidence is more abundant and robust. These include two viral (Human Metapneumovirus and Respiratory Syncytial Virus), one bacterial (pathogenic *E. coli*), and two parasitic (*Cryptosporidium* spp. and *G. duodenalis*) pathogens.

## 2. Example 1: Human Metapneumovirus and Respiratory Syncytial Virus

Human Metapneumovirus (HMPV) and Respiratory Syncytial Virus (HRSV) are single-stranded RNA viruses of the paramyxovirus family. In humans, HMPV infection is associated with acute respiratory tract disease (especially in young children) and can range from upper respiratory tract involvement to severe bronchiolitis and pneumonia. HMPV was first described in 2001 in paediatric patients with unidentifiable viruses in the Netherlands, although a concomitant serological survey evidenced that the virus had been circulating in humans for at least 50 years [78]. Age-related differences in clinical presentation have been described for HMPV infections: whereas children under two years of age were more frequently affected by cough, rhinitis, apnoea, and respiratory distress, those over two years of age were more likely to present crises, seizures, and fever [79].

In contrast, HRSV affects the lower respiratory tract in patients of all ages, although it is more frequent during childhood, being the primary cause of bronchiolitis and pneumonia in infants under one year of age [80]. HRSV (initially named Chimpanzee Coryza Agent, CCA) was first described as a respiratory disease characterised by coughing, sneezing, and runny nose in a colony of chimpanzees at an American medical research institution [81]. Subsequent investigations revealed that HRSV was also present in workers who had been in contact with infected chimpanzees and who were indeed the primary source of infection of the captive apes [81].

The dispersion of respiratory viruses is facilitated by the exhalation and transport of small droplets or aerosols produced by breathing, talking, coughing, or sneezing that can remain suspended in the air for long periods of time [82]. Airborne transmission is very effective among individuals sharing the same environment.

As humans have gradually come into closer proximity with wild great ape populations in recent decades, finding human-related HMPV and HRSV in habituated communities is not uncommon [40]. Indeed, respiratory disease is now considered the leading cause of death in human-habituated wild chimpanzees [20,83,84] and the second most common cause of death in mountain gorillas of all ages [85]. Most of the scientific evidence gathered so far has been obtained at Gombe National Park (GNP) in Tanzania and Taï National Park in Ivory Coast (chimpanzees) and the Virunga Volcanoes National Park (VVNP) in Rwanda, Uganda, and the Democratic Republic of the Congo (mountain gorillas). Although highly suspected, it is unclear whether the age-related disease patterns associated with HMPV and HRSV in human infections described above are also replicable in wild great ape populations.

Most deaths due to infectious diseases recorded among chimpanzees in the Kasakela community, GNP (Tanzania), have occurred during epidemics, with respiratory diseases being the most common. In total, respiratory diseases caused almost half of the deaths due to illness in this community between the years 1960 and 2006 [84].

In two other communities of GNP, sporadic outbreaks of respiratory diseases were described in the years 1996 and 2002, respectively [83]. The first outbreaks of respiratory disease in mountain gorillas were documented in the Rwandan part of the VVNP in 1988, when a flu-like illness was observed in three groups of gorillas habituated to tourism [86]. Subsequently, between 1990 and 2010, 18 new outbreaks were reported in wild mountain gorillas, also habituated to humans, from the same area. Worryingly, the number of respiratory disease outbreaks has been steadily increasing in frequency over time [87], paralleling the growing number of tourists visiting the park. Indeed, the visitors to VVNP increased by 82% between 2007 and 2016 [88].

Since the 1990s, signs of respiratory disease have also been observed in chimpanzees in Tanzania’s Mahale Mountains National Park (MMNP) [40,89,90]. Three large respiratory outbreaks were recorded in 2003, 2005, and 2006. Field investigations based on faecal samples and post-mortem tissues showed that the causative agent of the 2006 outbreak was a human-derived paramyxovirus, with cough and runny nose being identified as the predominant clinical signs in all cases. Weakness and lethargy were also frequently reported. In total, nine babies died, ranging in age from 2 months to 3 years old. Another nine chimpanzees (four infants, one juvenile, and four adults) have not been seen since the 2006 outbreak and may have died during the outbreak [40].

Between 2004 and 2006, four respiratory virus outbreaks were detected among chimpanzees at the Taï National Park, Ivory Coast. HMPV and HRSV were detected and identified by real-time and conventional PCR assays in faecal samples. It should be noted that faecal samples were used as surrogated biological samples since direct sampling of the respiratory tract was impractical for ethical and conservation reasons, whereas post-mortem tissues were only occasionally available. Comparative analyses revealed that the generated sequences were genetically identical to those from viruses previously isolated in lung tissue from chimpanzees that died due to respiratory disease [41].

In 2012 and 2013, a molecular epidemiological investigation was carried out at the VVNP during an ongoing outbreak of respiratory infections. The aim of the study was to test for the presence of HMPV and/or HRSV in faecal samples from human-habituated wild mountain gorillas and to assess the role of these viruses in the epidemic. HMPV and HRSV were detected by PCR in 12 of 20 samples from symptomatic gorillas but in none of the apparently healthy individuals (*n* = 81). Pathogenic human respiratory viruses were shown to be transmitted to gorillas and repeatedly introduced into mountain gorilla populations by people [39].

In 2014 and 2015, two respiratory virus outbreaks were detected in two habituated groups of bonobos in the province of Bandundu, Democratic Republic of the Congo [54]. In one of the groups (*n* = 40), at least four individuals died during the outbreak, and another seven individuals presented symptoms. In the other group (*n* = 30), at least 12 members presented symptoms, and 4 individuals died. HRSV was detected in both outbreaks, as well as co-infection with *Streptococcus pneumoniae* genetically identical to a type of human *S. pneumoniae* common in Africa [54].

Finally, in December 2016 and January 2017, simultaneous outbreaks of HMPV and human respirovirus 3 (HRV3) were detected in two chimpanzee communities in the same forest in Kibale, Uganda [34]. The viruses, which were absent before the outbreaks, each affected one of the chimpanzee communities, suggesting two independent episodes, but both of human origin, either directly or through intermediate hosts [34].

Although there have been very few studies carried out on wild orangutans, there is evidence showing that these great apes can contract respiratory diseases of human origin, both viral and bacterial [91].

## 3. Example 2: *Escherichia coli*

The common gastrointestinal bacterium *E. coli* is ubiquitous, can be transmitted through a variety of direct (via contact with infected animals, including humans) and indirect (via contaminated food, water, or fomites) routes, and has zoonotic potential [92]. Although generally benign, its pathogenic forms can cause disease, including diarrhoea, vomiting, and fever, among other symptoms [93,94,95]. Diarrhoeagenic *E. coli* can be classified into six main pathotypes, namely, enterotoxigenic *E. coli* (ETEC), enteropathogenic *E. coli* (EPEC), enteroaggregative *E. coli* (EAEC), Shiga-toxigenic *E. coli* (STEC), enteroinvasive *E. coli* (EIEC), and diffusely adherent *E. coli* (DAEC). Among them, STEC is regarded as a zoonotic pathogen of increasing public health concern [96,97].

Very few studies have addressed how often and by which routes *E. coli* is transmitted to wildlife in general (and great apes in particular). In addition, little or no information is available on the *E. coli* pathotypes infecting great apes or their drug resistance patterns [48].

Most of the studies investigating *E. coli* transmission from humans to great apes have been carried out at the Kibale and Bwindi Impenetrable National Parks (KNP and BINP, respectively) in Uganda, where some chimpanzee and gorilla communities have been intermittently surveyed for more than 30 years [47,98]. In a seminal study conducted at the KNP, ongoing research and ecotourism activities were identified as the main drivers enhancing contact between humans and great apes [47]. To assess whether human–ape interactions affect the diversity of the gut bacterial communities in these two host species, faecal samples from habituated wild chimpanzees and humans sharing a habitat were collected. Genetic analyses revealed that chimpanzees carried gut microbial signatures more similar to those of humans involved in research and ecotourism activities aimed at observing chimpanzees than to those from inhabitants of nearby villages who were not involved in those activities and had no contact with ape communities [47]. Additionally, the resistance to five antibiotics observed in *E. coli* isolates recovered from a chimpanzee naïve to drug treatment provided reasonable suspicion for the transmission of resistant *E. coli* strains or resistance-conferring genetic elements from humans to chimpanzees. These data also indicate that humans and apes interacting in the wild can share genetically and phenotypically similar gastrointestinal *E. coli* communities, presumably originating from common environmental sources [47].

Subsequent studies conducted in rural settings within the KNP and the BINP demonstrated that genetic profiles of gut bacteria communities isolated from humans and their cattle varied with geographic distance [99]. Humans harboured bacteria genetically more similar to those from cattle living in the same location than to bacteria from humans living in other, farther away locations. The same phenomenon was also true for cattle populations. Overall, this study suggested that close contact between people and their livestock could lead to high *E. coli* inter-species transmission rates. In fact, human–cattle bacterial genetic similarity within a locus has been demonstrated to be even greater than human–human or cattle–cattle bacterial genetic similarity between loci [99].

Another study conducted at the BINP showed that mountain gorillas (*Gorilla gorilla beringei*) carried *E. coli* communities genetically similar to those circulating in close-by human and cattle populations. In contrast, mountain gorillas that did not share a habitat with humans or their cattle harboured *E. coli* communities genetically distant to those typically present in humans and cattle [46]. Overall, these data indicated that the genetic dissimilarity of *E. coli* communities present in mountain gorillas and humans increased with geographical distance. However, no obvious trend was observed between the genetic structure of the *E. coli* communities present in mountain gorillas and cattle sharing their habitats. Drug resistance patterns observed at KNP and BINP also showed that 17% of *E. coli* isolates from mountain gorillas were resistant to at least one antibiotic used by the local human population. The prevalence of antibiotic-resistant *E. coli* strains was directly proportional to the degree of overlap of gorilla and human habitats [99].

Therefore, human–ape interactions may also favour the unnoticed spreading of multidrug-resistant bacteria, that is, bacteria that are not susceptible to at least one antimicrobial agent in three or more classes of antimicrobials [100]. These include beta-lactams, macrolides, fluoroquinolones, tetracycline, and aminoglycosides. These superbugs are regarded as good markers to assess the dynamics of bacterial transfer between humans and wildlife, as resistant strains are thought to be naturally selected in anthropogenic environments and then transferred to wildlife [101,102]. A study in national parks in Gabon found *E. coli* strains resistant to amoxicillin and other antibiotics in the faecal material of western lowland gorillas (*G. gorilla gorilla*). Since *E. coli* was not intrinsically resistant to these antibiotics, the authors speculated that those bacterial strains were acquired from humans [103].

In another study carried out in the Dzanga-Sangha Protected Area (Central African Republic), faecal samples from humans, habituated and non-habituated gorillas, and other wild animal species were cultured, and isolated enterobacteria strains carrying resistance genes were searched and compared [33]. Although multilocus sequence typing (MLST) and pulsed-field gel electrophoresis (PFGE) analyses could not confirm inter-species transmission, multiresistant *E. coli* isolates with the *qepA* gene (a plasmid-mediated gene responsible for reduced fluoroquinolone susceptibility) were found in a non-habituated gorilla. Since *qepA*-carrying multidrug-resistant plasmids are known to be generated primarily under selective antimicrobial pressure [103,104], the authors hypothesised that the *E. coli* isolate found in a gorilla may have had a human origin. This assumption was further supported by the finding of two *qepA E. coli* isolates from people living in nearby villages, although MLST and PFGE genotyping data from the human and gorilla isolates could not demonstrate any relationship between them [33]. Importantly, plasmid-mediated resistance genes and multiresistant isolates were found in both habituated and non-habituated gorillas. This fact strongly suggests that human proximity (i.e., tourists and researchers) is not the only contributing factor to the colonisation of the gorilla gut with resistant bacteria. Sharing habitats may be just as, or even more, important for microbiota transmission as close contact with people [33]. Since most studies investigating bacterial cross-transmission between humans and wild great apes were based on habituated ape populations (Table 1), it would be interesting to see whether the above-mentioned results were the product of a causal event or a true trend. More research, particularly tackling non-habituated animals, should be conducted to clarify this issue.

## 4. Example 3: *Cryptosporidium* spp. and *Giardia duodenalis*

*Cryptosporidium* spp. and *G. duodenalis* are widespread gastrointestinal protozoan parasites that affect a wide range of mammalian hosts [105,106,107]. Whereas *Cryptosporidium* spp. are obligate intracellular pathogens that affect the epithelium of the digestive and respiratory tracts [108], the vegetative form (trophozoite) of *G. duodenalis* is most often attached to the lining of the small intestine [109]. Both *Cryptosporidium* spp. and *G. duodenalis* are significant diarrhoea-causing pathogens affecting human and non-human primate populations, particularly in sub-Saharan Africa and Southeast Asia [63,110,111]. Transmission is via the faecal–oral route, either through direct animal–animal (including human) contact or indirectly through the ingestion of contaminated food and water [107,112,113].

A significant number of *Cryptosporidium* and *Giardia* species/genotypes are zoonotic [114,115]. Molecular typing of both parasites in human and animal reservoirs is, therefore, essential to unravel how often and to what extent anthroponotic, zoonotic, and spillback transmission cycles contribute to the spreading of giardiasis and cryptosporidiosis. However, *Cryptosporidium* and *G. duodenalis* detection and typing are challenging in wild non-human primates (particularly great apes) due to typically low population numbers, difficult access to natural habitats, limited range of biological specimens for testing, and conservation and ethical issues. Consequently, available molecular typing information in those hosts remains scarce and patchy at both spatial and temporal scales.

A seminal study conducted at the BINP in Uganda revealed that park staff members interacting with mountain gorillas shared with them the very same *Cryptosporidium* genetic variant at a much higher proportion (21%) than local community people who did not enter the park (3%) [68]. A subsequent survey in the same country found that *Cryptosporidium* spp. and *G. duodenalis* were relatively common (4–6%) in the faecal material of other non-human primates, such as colobus and guenon monkeys living in fragmented and degraded forests in Uganda, but were absent in their counterparts living in undisturbed forests [116]. Furthermore, semi-free bonobos (*Pan paniscus*), but not bonobos reintroduced into the wild, were found to be infected with *G. duodenalis* in the Democratic Republic of the Congo [37]. Taken together, these studies strongly suggest that anthropic-driven habitat disturbance and close human–animal contact increase the likelihood of *Cryptosporidium* spp. and *G. duodenalis* inter-species transmission and persistence in wild non-human primates [37,68,116].

The strongest evidence supporting the anthropic transmission of protozoan parasites to non-human primates (including great apes) comes from a molecular epidemiological study conducted in the Greater Gombe Ecosystem, Tanzania [65]. The study site comprised a rural area characterised by high rates of overlap among humans, domesticated animals, and wildlife. Remarkably, *Cryptosporidium hominis* (the most prevalent *Cryptosporidium* species circulating in humans globally) was found in 4.3% (8/185) of humans, 11.9% (10/84) of chimpanzees (*Pan troglodytes schweinfurthii*), and 10.6% (5/47) of baboons (*Papio anubis*) investigated. All *C. hominis* isolates successfully genotyped at the 60 kDa glycoprotein (*gp60*) marker (seven in humans, five in chimpanzees, and three in baboons) were identified as subtype IfA12G2. It should be noted that the *gp60* subtype family If, although less common than Ia, Ib, Id, and Ie, has been previously described in human cases of cryptosporidiosis in Kenya, South Africa, and Tanzania [111]. Infected local human populations were suspected of serving as the source of infections with *C. hominis* IfA12G2 in chimpanzees and baboons.

*Cryptosporidium hominis* has also been reported at low (<5%) infection rates in subsequent studies conducted in other sub-Saharan African countries. For instance, *C. hominis* was identified in 2.6% (6/235) of recently captured olive baboons in rural or forest areas of Kenya [117]. Genotyping analyses at the *gp60* marker allowed the identification of subtype families Ib, If, and Ii. Whereas the subtype family Ii (together with Ik, Im, and In) is widely thought to be animal-adapted and rarely or not at all seen in humans [106], that is not the case for members of subtype families Ib and If, both prevalent in African human populations [111]. Judging by the common occurrence of *gp60* subtype family Ib in humans in Kenya and the close contact of baboons with humans, it is reasonable to hypothesise that baboons might have acquired the *C. hominis* infections from local community people [117].

Similarly, *C. hominis* has also been found in 4% (1/25) of lemurs in Madagascar [118] and in <1% of wild chimpanzees (*Pan troglodytes verus*) in Senegal (0.4%, 1/235) [60] and Ivory Coast (0.8%, 1/124) [59]. A lack of genotyping data, or failure to generate them, makes it difficult to assess the occurrence and directionality of potential transmission events between human and non-human primate hosts. However, low *C. hominis* prevalences and human–chimpanzee encounter rates in Senegal and Ivory Coast seem to indicate that poachers, rangers, or field researchers were the most likely source of these infections.

Genotypes (assemblages) A and B are the dominant *G. duodenalis* genetic variants circulating in human and non-human primates globally [111,119]. An early molecular typing study conducted at the BINP in Uganda found that *Giardia*-positive free-ranging human-habituated mountain gorillas and humans (park staff, local community people, soldiers, and tourists) with varying degrees of contact with them were all infected by the very same assemblage A genetic variant [70]. It was presumed that this genotype might have been introduced into gorilla populations through habituation and/or ecotourism activities and subsequently sustained in their habitats by anthropozoonotic transmission. Uncontrolled human defecation within the park may have contributed to environmental contamination and parasite spreading [70]. Additionally, *G. duodenalis* assemblage A, sub-assemblage AII was detected in wild western lowland gorillas that were under habituation and fully habituated (but not in unhabituated animals) in Dzanga-Sangha Protected Areas, Central African Republic [71]. Zoonotic *G. duodenalis* sub-assemblages BIII (*n* = 1) and BIV (*n* = 5) were also identified in mountain gorillas at the VVNP [61]. These wild great ape populations are facing intense ecological pressures due to their proximity to local community people and human-driven activities within the park, such as research, tourism, illegal hunting, and anti-poaching patrols. Unfortunately, no human populations were investigated in the above-mentioned surveys, so it is unclear to what extent the origin of the AII/BIII/BIV infections was of anthroponotic nature.

Although most of the research on *Cryptosporidium* spp. and *G. duodenalis* has been conducted in African wild great apes, few surveys have also been aimed at investigating their presence in orangutans, whose species are native to the rainforests of Borneo and Sumatra (Indonesia). In an early study investigating Bornean orangutans (*P. pygmaeus*) with different degrees of habituation to humans, *Giardia* spp. was identified in 3 out of 119 captive orangutans, but not in any of the semi-captive (*n* = 44) or wild (*n* = 61) individuals examined [120]. Similar results were observed in a subsequent survey carried out in orangutans (*n* = 298) from Borneo (*P. pygmaeus*) and Sumatra (*P. abelii*) [64]. In that study, zoonotic *C. parvum* (*n* = 2) was significantly less frequently detected in wild orangutans than in semi-wild or captive individuals, whereas *G. duodenalis* assemblage B (*n* = 1) was detected in a semi-captive orangutan with limited human exposure. Mimicking data generated in the African studies, human exposure increased the risk of infection with *Cryptosporidium* and *G. duodenalis* in orangutans. Orangutans, which are semi-solitary by nature, when kept in semi-liberty or in captivity, live in unnatural conditions that can make them more susceptible to anthropic pathogens [64,120].

## 5. Gaps in Knowledge and Future Perspectives

Current research on anthroponotic pathogen transmission in wild great apes is hampered by several drawbacks:

Three out of four (76%) of the forty-three studies considered in Table 1 were carried out in habituated populations, 12% were carried out in both habituated and non-habituated populations, and only 7% were carried out in non-habituated populations (this status was unknown for the remaining 5% of the surveys). Therefore, there is an obvious lack of knowledge on human-borne diseases affecting wild non-habituated great apes, which are precisely the most unprotected communities.There is a clear bias on geographical representativeness, as 12 studies (28%) were conducted in Uganda: 7 in habituated gorillas, 3 in habituated chimpanzees, 3 in both habituated and non-habituated gorillas, and none in non-habituated great apes (Table 1), while there are no studies in several chimpanzee- and gorilla-harbouring countries, such as Nigeria, Equatorial Guinea, Liberia, Sierra Leona, and Guinea Conakry.Most available studies tend to focus on the very same ape communities over time. This is the case for Ivory Coast, where virtually all investigations on infectious diseases affecting chimpanzees have been carried out in habituated groups at the Taï National Park (Table 1). Only a single study was conducted on non-habituated chimpanzees at the Comoé National Park [59].At the global level, it is unknown what percentage of the current great ape communities has been studied, since obtaining a reliable population census (in both Africa and Asia) is a hard task (Appendix A). This difficulty is accentuated in countries with greater political instability, such as the Democratic Republic of the Congo, whose forests represent half of the total area of tropical rainforest in Africa and provide shelter for two species of gorilla, two subspecies of the common chimpanzee, and the bonobo. The Democratic Republic of the Congo has wanted to imitate the very lucrative ecotourism industry developed in Rwanda and, to a lesser extent, in Uganda, but its instability is a major obstacle to successfully achieving this goal [121]. This instability translates into a scarcity of studies on the welfare of gorillas and chimpanzees due to the intrinsic complications involved in working under these conditions.Ideally, species conservation should be based on good knowledge of the population size of the species and the threats to its survival (such as infectious diseases) throughout its geographic range [1]. Therefore, it is very difficult to estimate the extent to which human-borne diseases pose an actual threat to wild great ape populations when we do not know what we are trying to protect.Direct sampling for diagnostic testing is not possible except under very specific circumstances in habituated animals (e.g., chemical immobilisation for clinical or surgical treatment). Non-invasively collected faecal samples often remain the only alternative available [39]. However, faecal material may not be optimal for the detection of several pathogens. Additionally, fresh faecal samples from non-habituated apes may be difficult to obtain since they are collected opportunistically.

Ideally, future research studies aiming to investigate human–ape disease transmission should be directed to cover the following aspects:

Improving the accuracy and reliability of current estimates of abundance, geographical distribution, and existing habituated and non-habituated great ape groups, both at national and international levels.Providing a better account of the actual burden of infectious diseases in habituated animals within the total population and assessing the potential risk of disease transmission from habituated apes to unhabituated ones.Improving networking capabilities among research groups working in the field, so generated information (e.g., census data) can be deposited in publicly accessible repositories. At present, information generated by small research investigations in locally conducted projects does not always reach the general scientific community.Molecular typing surveys at the sub-genotype/strain level are greatly needed. Ideally, such studies should be carried out under the One Health umbrella and include human, animal, and environmental (surface water, pastures, or green leaves) samples. Under the appropriate design, this information is essential to characterise transmission dynamics and accurately demonstrate the occurrence and directionality of zoonotic events.Research should be expanded to non-habituated great ape communities, as these populations typically inhabit the most encroached environments and are therefore more vulnerable to environmental threats, including human-borne infectious diseases.

## Figures and Tables

**Table 1 vetsci-09-00356-t001:** Pathogens of viral, bacterial, parasitic, and fungal nature for which reverse zoonotic transmission from humans to wild great apes is suspected.

Pathogenic Agent	Human Origin?	Affected Great Ape Species	Human Contact?	Country(ies)	Reference
**Viruses**					
Ebola virus	Unlikely	G.g., P.t.	H	Congo, Gabon	[35]
Human adenovirus A–F	Probable	G.g., P.p., P.t.	H	Cameroon, Ivory Coast, Democratic Republic of Congo, Gambia, Republic of Congo, Rwanda, Tanzania, Uganda	[36]
Human adenoviruses B, C, E	Probable	P.p.	H	Democratic Republic of Congo	[37]
Human coronavirus OC43	Probable	P.t.v.	H	Ivory Coast	[38]
Human metapneumovirus	Probable	G.g.b.	H	Rwanda	[39]
	Probable	P.t.	H	Tanzania	[40]
	Probable	P.t.v.	H	Ivory Coast	[41]
	Probable	P.t.s.	H	Uganda	[34]
Monkeypox virus	Uncertain	P.t.v.	H	Ivory Coast	[42]
Severe acute respiratory syndrome coronavirus 2	Probable	G.g., P.p., P.t., P.a.	Unknown	Africa (not specified)	[43]
T-cell lymphotropic virus 1–4	Probable	P.p.	H	Democratic Republic of Congo	[44]
**Bacteria**					
*Campylobacter* spp.	Uncertain	G.g.b.	H	Uganda	[45]
*Escherichia coli*	Probable	G.g.b.	H, NH	Uganda	[46]
	Uncertain	G.g.g.	H, NH	Central African Republic	[33]
	Probable	P.t.	H	Uganda	[47]
	Uncertain	P.t.	H	Uganda	[48]
*Enterobacter* sp.	Unlikely	P.p.w.	H	Indonesia	[49]
*Klebsiella pneumoniae*	Unlikely	G.g.b.	H	Rwanda	[50]
	Uncertain	G.g.g.	H, NH	Central African Republic	[33]
	Unlikely	P.p.w.	H	Indonesia	[49]
*Mycobacterium leprae*	Unlikely	P.t.v.	H, NH	Ivory Coast, Guinea Bissau	[51]
*Mycobacterium tuberculosis*	Unlikely	P.t.v.	H	Ivory Coast	[52]
	Unlikely	Great apes	Unknown	Africa (several countries)	[53]
*Pseudomonas aeruginosa*	Unlikely	P.p.w.	H	Indonesia	[49]
*Salmonella* spp.	Uncertain	G.g.b.	H	Uganda	[45]
	Uncertain	P.t.	H	Uganda	[48]
*Shigella* spp.	Uncertain	G.g.b.	H	Uganda	[45]
	Uncertain	P.t.	H	Uganda	[48]
*Staphylococcus aureus*	Uncertain	P.p.w.	H	Indonesia	[49]
*Streptococcus pneumoniae*	Uncertain	G.g.b.	H	Rwanda	[50]
	Probable	P.p.	H	Democratic Republic of Congo	[54]
	Uncertain	P.t.v.	H	Ivory Coast	[55]
	Probable	P.t.v.	H	Ivory Coast	[56]
*Treponema pallidum pertenue*	Uncertain	G.g.g.	NH	Republic of Congo	[57]
**Parasites (microeukaryotes)**			H		
*Balantioides coli*	Unlikely	P.t.s.	H	Tanzania	[58]
*Blastocystis* sp.	Probable	P.t.v.	NH	Ivory Coast	[59]
	Probable	P.t.v.	NH	Senegal	[60]
*Cryptosporidium* spp.	Uncertain	G.g.b.	H	Rwanda	[61]
	Uncertain	G.g.b.	H, NH	Uganda	[62]
	Uncertain	G.g.b.	H, NH	Uganda	[63]
	Uncertain	P.a., P.pyg.	H	Indonesia	[64]
*Cryptosporidium hominis*	Probable	P.t.s.	H	Tanzania	[65]
	Probable	P.t.v.	NH	Ivory Coast	[59]
	Probable	P.t.v.	NH	Senegal	[60]
*Cryptosporidium meleagridis*	Unlikely	G.g.b.	H	Rwanda	[66]
*Cryptosporidium muris*	Unlikely	G.g.b.	H	Rwanda	[66]
*Cryptosporidium parvum*	Uncertain	G.g.b.	H	Uganda	[67]
	Uncertain	G.g.b.	H	Uganda	[68]
*Entamoeba histolytica*	Very probable	P.t.s.	H	Tanzania	[58]
	Very probable	P.a., P.pyg.	H	Indonesia	[69]
*Giardia duodenalis*	Probable	G.g.b.	H	Rwanda	[61]
	Probable	G.g.b.	H, NH	Uganda	[62]
	Probable	G.g.b.	H	Uganda	[70]
	Probable	G.g.g.	H	Central African Republic	[71]
	Probable	P.p.	H	Democratic Republic of Congo	[37]
	Probable	P.t.v.	NH	Ivory Coast	[59]
	Probable	P.t.v.	NH	Senegal	[60]
	Probable	P.a., P.pyg.	H	Indonesia	[64]
*Iodamoeba bütschlii*	Probable	P.t.s.	H	Tanzania	[58]
*Plasmodium ovale wallikeri*	Uncertain	G.g.g.	H	Central African Republic	[72]
**Parasites (helminths)**			H		
*Abbreviata caucasica*	Probable	P.t.s.	H	Tanzania	[58]
*Ascaris* sp.	Very probable	P.t.s.	H	Tanzania	[58]
*Bertiella* sp.	Unlikely	P.t.s.	H	Tanzania	[58]
*Capillaria hepatica*	Probable	G.g.b.	H	Rwanda	[73]
Dicrocoeliidae	Probable	P.t.s.	H	Tanzania	[58]
*Trichuris* sp.	Probable	P.t.s.	H	Tanzania	[58]
*Necator americanus*	Probable	G.g.g.	H	Central African Republic	[74]
*Oesophagostomum* sp.	Probable	P.t.s.	H	Tanzania	[58]
*Oesophagostomum bifurcum*-like	Unlikely	P.p.	H	Democratic Republic of Congo	[37]
*Oesophagostomum stephanostomum*	Unlikely	P.p.	H	Democratic Republic of Congo	[37]
*Strongyloides fulleborni*	Unlikely	P.t.s.	H	Tanzania	[58]
*Strongyloides stercoralis*	Probable	P.p.	H	Democratic Republic of Congo	[37]
*Taenia solium*	Unlikely	P.p.	H	Democratic Republic of Congo	[37]
**Parasites (arthropods)**					
*Sarcoptes scabiei*	Probable	G.g.b.	H	Uganda	[75]
	Probable	G.g.b.	H	Uganda	[76]
**Fungi**			H		
*Enterocytozoon bieneusi*	Probable	G.g.b.	H	Rwanda	[66]
*Encephalitozoon cuniculi*	Probable	G.g.b.	H	Rwanda	[66]
*Encephalitozoon intestinalis*	Probable	G.g.b.	H	Uganda	[77]

G.g., *Gorilla gorilla*; G.g.b., *Gorilla gorilla beringei*; G.g.g., *Gorilla gorilla gorilla*; H: habituated to human presence; NH: non-habituated to human presence; P.a., *Pongo abelii*; P.p., *Pan paniscus*; P.pyg., *Pongo pygmaeus*; P.p.w., *Pongo pygmaeus wurmbii*; P.t., *Pan troglodytes*; P.t.s., *Pan troglodytes schweinfurthii*; P.t.v., *Pan troglodytes verus*.

## Data Availability

Not applicable.

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
