# Peer review of "Human-Borne Pathogens: Are They Threatening Wild Great Ape Populations?"

_vetsci, 2022, doi:10.3390/vetsci9070356_

Round 1
Reviewer 1 Report
The manuscript entitled 'Human-borne pathogens: Are they threatening wild great ape populations?' is very well-written and present current information on human-borne pathogens with the possibility and probable link to be transferred to the wild great ape populations.
Minor comments:
Line 43-44: Authors used generic terms for the specific animal species in line 43 (eastern and western gorilla). In the next line 44, the authors used a scientific name. Use either a generic name or a scientific name.
Line 119: The first sentence is incomplete. Re-write the sentence.
Table 1: Correct the name of the second virus as Human adenoviruses A-F.
Table S1: Use ‘Country’ as the headline for column 1.
Table S1: Make the number ‘339’ in the same alignment with others in column 3.
Author Response
Referee #1
The manuscript entitled 'Human-borne pathogens: Are they threatening wild great ape populations?' is very well-written and present current information on human-borne pathogens with the possibility and probable link to be transferred to the wild great ape populations.
Reply: we thank the preliminary positive appraisal by Reviewer #1.
- Line 43-44: Authors used generic terms for the specific animal species in line 43 (eastern and western gorilla). In the next line 44, the authors used a scientific name. Use either a generic name or a scientific name.
Reply: please note that great ape scientific (in full) and common names were provided the first time that they were mentioned in the text. Afterwards, common or abbreviated names were used indistinctly depending on the context of the sentence to make the reading easier and more fluid. For instance, in line 58 we refer to eastern and western gorilla populations, whose scientific names were already provided in lines 54 and 55.
- Line 119: The first sentence is incomplete. Re-write the sentence.
Reply: the sentence has been rephrased as “Close and frequent contact with humans is not the only factor contributing to disease transmission to great apes” to improve clarity and readability.
- Table 1: Correct the name of the second virus as Human adenoviruses A-F.
Reply: Please note that the name stated in Table 1 is correct, as this study does not include human adenoviruses A, D, or F.
- Table S1: Use ‘Country’ as the headline for column 1.
Reply: Thanks for noticing this mistake. Added as per requested.
- Table S1: Make the number ‘339’ in the same alignment with others in column 3.
Reply: Please note that this figure corresponds to both Gorilla beringei and Gorilla gorilla species, as no data disaggregated by individual species were available. This is the reason of presenting the figure in the middle of the two columns as a merged cell.
Reviewer 2 Report
Review: Human-borne pathogens: Are they threatening wild great ape populations?
This is an interesting and potentially important manuscript about anthroponotic pathogen infection in wild great apes. The manuscript is well written. However, I would suggest some minor modifications.
The paper does not include any schematic figures, including a graphic abstract. I would appreciate it if you could include a graphic abstract or a graphic figure for each of the examples.
Author Response
Referee #2
This is an interesting and potentially important manuscript about anthroponotic pathogen infection in wild great apes. The manuscript is well written. However, I would suggest some minor modifications.
Reply: we thank the preliminary positive appraisal by Reviewer #2.
The paper does not include any schematic figures, including a graphic abstract. I would appreciate it if you could include a graphic abstract or a graphic figure for each of the examples.
Reply: Please note that providing a graphical abstract is not compulsory for Veterinary Sciences, so we feel adding such figure would not add value to the review besides the aesthetic change.
Reviewer 3 Report
This review article summarized current information on the most relevant pathogens of viral, bacterial, parasitic, and fungal nature for which transmission from humans to wild great apes is suspected. In my opinion the review is well written and table is clearly presented. I recommend this manuscript for acceptance.
Author Response
Referee #3
This review article summarized current information on the most relevant pathogens of viral, bacterial, parasitic, and fungal nature for which transmission from humans to wild great apes is suspected. In my opinion the review is well written, and table is clearly presented. I recommend this manuscript for acceptance.
Reply: we thank the preliminary positive appraisal by Reviewer #3.